# BayesOpt Adversarial Attack

**Binxin Ru, Adam D. Cobb, Arno Blaas**
Machine Learning Research Group,
Department of Engineering Science,
University of Oxford
{robin, acobb, arno}@robots.ox.ac.uk

**Yarin Gal**
OATML Research Group,
Department of Computer Science,
University of Oxford
yarin.gal@cs.ox.ac.uk

## Abstract

Black-box adversarial attacks require a large number of attempts before finding successful adversarial examples that are visually indistinguishable from the original input. Current approaches relying on substitute model training, gradient estimation or genetic algorithms often require an excessive number of queries. Therefore, they are not suitable for real-world systems where the maximum query number is limited due to cost. We propose a query-efficient black-box attack which uses Bayesian optimisation in combination with Bayesian model selection to optimise over the adversarial perturbation and the optimal degree of search space dimension reduction. We demonstrate empirically that our method [1] can achieve comparable success rates with 2-5 times fewer queries compared to previous state-of-the-art black-box attacks.

## 1 Introduction

Deep learning algorithms are widely deployed in many real-world systems and are increasingly being used for tasks, ranging from identity verification (Liu et al., 2018), to financial services (Heaton et al., 2017) to autonomous driving (Bojarski et al., 2016). However, even the most accurate deep learning models can be easily deceived by perturbations which are visually imperceptible to the human eye (Szegedy et al., 2013; Carlini et al., 2016). The growing costs and risks associated with the potential model failures has led to the importance of studying adversarial attacks, both in assessing their robustness and their ability to detect such attacks. In this paper we focus on highly practical adversarial attacks that fulfill the following two criteria. First, the attack is designed for a black-box setting because in real-world examples, the attacker would normally have no knowledge of the target deep learning model and can only interact with the model by querying it. Second, query efficiency is highly prioritised because in practical cases where the damage caused by the attack is high, the query budget available to the attacker will be highly limited due to the risk of being detected by the defence system or other high inherent costs (monetary or computational) of model evaluations.

Despite of the large array of adversarial attacks proposed in the literature, many of them are white-box approaches that assume full access to the target model architecture and the ability of performing back-propagation to get gradient information (Moosavi-Dezfooli et al., 2016; Kurakin et al., 2016; Gu & Rigazio, 2014; Goodfellow et al., 2014; Chen et al., 2018; Carlini & Wagner, 2017). On the other hand, for black-box attacks, there are various techniques that have been used which do not require access to the model architecture. One class of methods trains a white-box substitute model and attacks the target model with adversarial examples that successfully fool the substitute (Papernot et al., 2017). However, this type of method requires the availability of the original training data or large query data to train the substitute network and the performance is often limited by the mismatch between the substitute and the target models (Su et al., 2018). The second class of black-box attacks, which show better empirical performance than the substitute model approaches, numerically estimate the gradient of the target model by repeatedly querying it (Chen et al., 2017; Ilyas et al., 2018; Tu et al., 2018) and attack with the estimated gradient. Although various techniques are employed to increase the query efficiency for the gradient estimation, they need an excessively large query budget to achieve a successful attack (Alzantot et al., 2018). Another line of work removes the need for gradient estimation and uses decision-based techniques (Brendel et al., 2017) or genetic algorithms

---

[1]Our code is available at https://github.com/rubinxin/BayesOpt_Attack.git

(Alzantot et al., 2018) to generate adversarial examples. One popular technique that is adopted by many black-box attacks (Chen et al., 2017; Alzantot et al., 2018; Tu et al., 2018) to significantly improve the query efficiency is to search for adversarial perturbations in a low-dimensional latent space (search dimensionality reduction). However, learning the effective dimensionality of the latent search space can be challenging by itself, and has not been investigated by the prior works to the best of our knowledge.

In light of the above limitations, we propose a query efficient black-box attack that iteratively optimises over both the adversarial perturbation and the effective dimensionality of the latent search space. Our main contributions are summarised as follows:

- We introduce a novel gradient-free black-box attack method, BayesOpt attack, which uses Bayesian optimisation with Gaussian process surrogate models to find the effective adversarial example and is capable of dealing with high-dimensional image inputs.

- We proposes a Bayesian technique which learns the optimal degree of search dimensionality reduction by harnessing our statistical surrogate and information from query data. This technique can be incorporated naturally into our attack procedure, leading to efficient optimisation over both adversarial perturbation and the latent search space dimension.

- We empirically demonstrate that under the $L_\infty$ constraint, our proposed attack method can achieve comparable success rate with about *2 to 5 times* fewer model queries in comparison to current state-of-the-art query-efficient black box attack methods.

## 2 RELATED WORK ON BLACK-BOX ATTACKS

Most of the existing adversarial attacks (Moosavi-Dezfooli et al., 2016; Kurakin et al., 2016; Gu & Rigazio, 2014; Goodfellow et al., 2014; Chen et al., 2018; Carlini & Wagner, 2017) focus on white-box settings, where the attacker can get full access to the target model and has complete knowledge of the architecture, weights and gradients to generate successful adversarial examples. However, real-world systems are usually attacked in a black-box setting, where one has no knowledge about the model and can only observe the input-output correspondences by querying the model. Here we give a brief overview of various existing black-box attacks.

**Substitute model** One class of black-box attacks uses data acquired from querying the target black-box model to train a substitute model (Papernot et al., 2017), which mimics the classification behaviour of the target model. The adversary can then employ any white-box method to attack the fully observable substitute model and apply the successful adversarial example to the target model. Such approaches rely on the transferability assumption that adversarial examples which are effective to the substitute model are also very likely to conceive the target model given their similar classification performance on the same data (Szegedy et al., 2013). Moreover, to provide training data for the substitute model, the adversary either requires information on the target model's training set, which is highly unrealistic in real-world applications, or needs to build a synthetic training set by querying the target model (Papernot et al., 2017), which implies a large number of model evaluations and becomes hardly feasible for large models or complex datasets (Brendel et al., 2017).

**Gradient estimation** An alternative to training a substitute model is to estimate the gradient via finite differences and use the estimated gradient information to produce attacks. However, the naive coordinate-wise gradient estimation requires excessive queries to the target model (2 queries per coordinate per descent/attack step) and thus is not feasible for attacking models with high dimensional inputs (e.g. classifiers on ImageNet). Chen et al. (2017) overcome this limitation by using stochastic coordinate gradient descent and selecting the batch of coordinates via importance sampling, introducing the state-of-the-art zeroth-order attack, ZOO, which achieves comparable attack success rate and perturbation costs as many white-box attacks. Although ZOO makes it computationally tractable to perform black-box attack on high-dimensional image data, it still requires millions of queries to generate a successful adversarial example, making it impracticable for attacking real-world systems where model query can be expensive and the budget limited. Improving on ZOO, AutoZOOM (Tu et al., 2018) significantly enhances the query efficiency by using random vectors to estimate the full gradient and adjusting the estimation parameter adaptively to trade off query efficiency vs. input perturbation cost. More importantly, AutoZOOM shows the benefits of employing dimension reduction techniques in accelerating attacks (i.e. searching for adversarial perturbation in a low-dimensional

latent space and decoding it back to the high-dimensional input space). Parallel work by Ilyas et al. (2018) estimate the gradient through a modified version of natural evolution strategy which can be viewed as a finite-difference method over a random Gaussian basis. The estimated gradient is then used with projected gradient descent, a white-box attack method, to generate adversarial examples.

**Gradient-free optimisation** As discussed, gradient-estimation approaches in general need an excessive number of queries to achieve successful attack. Moreover, their dependence on the gradient information makes them less robust to defences which manipulate the gradients (Athalye et al., 2018; Brendel et al., 2017; Guo et al., 2017). Thus, truly gradient-free methods are more likely to bypass such defences. One recent example, which has demonstrated state-of-the-art query efficiency, is GenAttack (Alzantot et al., 2018). GenAttack uses genetic algorithms to iteratively evolve a population of candidate adversarial examples. Besides having an annealing scheme for mutation rate and range, GenAttack also adopts dimensional reduction techniques, similar to AutoZOOM, to improve the query efficiency. In parallel to GenAttack, Brendel et al. (2017) introduce a decision-based attack, Boundary Attack, which only requires access to the final model decision. Boundary Attack starts from a huge adversarial perturbation and then iteratively reduces the perturbation through a random walk along the decision boundary. However, Boundary Attack takes about 72 times more queries than GenAttack to fool an undefended ImageNet model (Alzantot et al., 2018). Another recent approach introduced in Moon et al. (2019) reduces the search space to a discrete domain and subsequently uses combinatorial optimisation to find successful attacks, mostly focusing on the less challenging setting of untargeted attacks. Finally, the prior works that use Bayesian optimisation for adversarial attacks (Suya et al., 2017; Zhao et al., 2019) only investigate the use of simple Gaussian process as the surrogate model and are limited in their performance. The method proposed in (Suya et al., 2017) deals with the untargetted attack setting and only demonstrates the effectiveness of Bayesian optimisation in comparison to random search on a low-dimensional ($d = 57$) email attack task. BO-ADMM proposed in (Zhao et al., 2019) works on image data but it applies Bayesian optimisation directly on the search space of image dimension to minimise the joint objective of attack loss and distortion loss. Despite its query efficiency, BO-ADMM leads to poor-quality adversarial examples of large distortion loss.

## 3 PRELIMINARIES

### 3.1 PROBLEM DEFINITION

We focus on the black-box attack setting, where the adversary has no knowledge about the network architecture, weights, gradient or training data of the target model $f$, and can only query the target model with an input $\boldsymbol{x}$ to observe its prediction scores on all $C$ classes (i.e. $f : \mathbb{R}^d \longrightarrow [0,1]^C$) (Tu et al., 2018; Alzantot et al., 2018). Moreover, we aim to perform *targeted* attacks, which is more challenging than untargeted attacks, subject to a constraint on the maximum change to any of the coordinates (i.e., a $L_\infty$ constraint) (Warde-Farley & Goodfellow, 2016; Alzantot et al., 2018). Specifically, *targeted* attacks refer to the case where given a valid input $\boldsymbol{x}_{origin}$ of class $t_{origin}$ (i.e. $\arg\max_{i\in\{1,...,C\}} f(\boldsymbol{x}_{origin})_i = t_{origin}$) and a target $t \neq t_{origin}$, we aim to find an adversarial input $\boldsymbol{x}$, which is close to $\boldsymbol{x}_{origin}$ according to the $L_\infty-$norm, such that $\arg\max_{i\in\{1,...,C\}} f(\boldsymbol{x})_i = t$. *Untargeted* adversarial attacks refer to the case that instead of classifying $\boldsymbol{x}_{origin}$ as $t_{origin}$, we try to find an input $x$ so that $\arg\max_{i\in\{1,...,C\}} f(\boldsymbol{x})_i \neq t_{origin}$.

In our approach, we follow the convention to optimise over the perturbation $\boldsymbol{\delta}$ instead of the adversarial example $\boldsymbol{x}$ directly (Chen et al., 2017; Alzantot et al., 2018; Tu et al., 2018). Therefore, our problem can be formulated as:

$$\arg\max_{i\in\{1,...,C\}} f(\boldsymbol{x}_{origin} + \boldsymbol{\delta})_i = t \quad \text{s.t.} \quad \|\boldsymbol{\delta}\|_\infty \leq \delta_{max} \tag{1}$$

### 3.2 BAYESIAN OPTIMISATION

Bayesian optimisation (BayesOpt) is a query-efficient approach to tackle global optimisation problems (Brochu et al., 2010). It is particularly useful when the objective function is a black-box and is costly to evaluate. There are 2 key components in BO: a statistical surrogate, such as a Gaussian process (GP) or a Bayesian neural network (BNN) which models the unknown objective, and an

acquisition function $\alpha(\cdot)$ which is maximised to recommend the next query location by trading off exploitation and exploration (see Algorithm 3 in Appendix A for standard BayesOpt).

## 4 BAYESOPT ATTACK

### 4.1 BAYESOPT ATTACK OBJECTIVE

It has been shown that reducing the dimensionality of the search space in Equation (1) increases query efficiency significantly (Chen et al., 2017; Tu et al., 2018; Alzantot et al., 2018). Due to our focus on the small query regime where our surrogate model needs to be trained with a very small number of observation data, we adopt the previously suggested dimensionality reduction technique, bilinear resizing, to reduce the challenging problem of optimising over high-dimensional input space of $x \in \mathbb{R}^d$ to one over a relatively low-dimensional input space, setting $x = x_{origin} + g(\delta)$ where $\delta \in \mathbb{R}^{d^r}$ with $d^r < d$ and $g : \mathbb{R}^{d^r} \longrightarrow \mathbb{R}^d$ being the bilinear resizing decoder[2].

Furthermore, we follow the approach of smoothing the discontinuous objective function in Equation (1), which has been found to be beneficial in previous work (Chen et al., 2017; Tu et al., 2018; Alzantot et al., 2018). Together with the dimensionality reduction, this leads to the following black-box objective problem for our Bayesian optimisation:

$$\delta^* = \arg\max_{\delta} y(\delta) \quad \text{s.t.} \quad \delta \in [-\delta_{max}, \delta_{max}]^{d^r} \tag{2}$$

$$\text{where} \quad y(\delta) = \left[ \log f(x_{origin} + g(\delta))_t - \log \sum_{j \neq t}^{C} f(x_{origin} + g(\delta))_j \right]$$

### 4.2 GP-BASED BAYESOPT ATTACK

We first use BayesOpt with a standard GP as the surrogate to solve for the black-box attack objective in the reduced input dimension in Equation (2). The GP encodes our prior belief on the objective $y$ :

$$y \sim \mathcal{GP}(\mu(\delta), k(\delta, \delta')) \tag{3}$$

which is specified by a mean function $\mu$ and a kernel/covariance function $k$ (we use the Matern-5/2 kernel in our work). In our work, we normalise the objective function value and thus use a zero-mean prior $\mu(\cdot) = 0$. The predictive posterior distribution for $y_t$ at a test point $\delta_t$ conditioned on the observation data $\mathcal{D}_{t-1} = \{\delta_i, y_i\}_{i=1}^{t-1} = \{\delta_{1:t-1}, y_{1:t-1}\}$ then has the form:

$$p(y_t|\delta_t, \mathcal{D}_{t-1}) = \mathcal{GP}(y; \mu_y(\cdot), k_y(\cdot, \cdot)) \tag{4}$$

where

$$\mu_y(\delta_t) = k(\delta_t, \delta_{1:t-1}) K_{1:t-1}^{-1} y_{1:t-1}, \tag{5}$$

$$k_y(\delta_t, \delta_t') = k(\delta_t, \delta_t') - k(\delta_t, \delta_{1:t-1}) K_{1:t-1}^{-1} k(\delta_{1:t-1}, \delta_t'). \tag{6}$$

where $\mathbf{K}_{1:t-1} = \mathbf{K}(\delta_{1:t-1}, \delta_{1:t-1})$ is the $t-1 \times t-1$ matrix with pairwise covariances $k(\delta_l, \delta_m')$, $l, m \leq t-1$ as entries. The optimal GP hyper-parameters $\theta^*$ such as the length scales and variance of the kernel function $k$ can be learnt by maximising the marginal likelihood $p(\mathcal{D}_{t-1}|\theta)$ which has analytic form as presented in Appendix F. Please refer to (Rasmussen, 2003) for GPs.

Based on the predictive posterior distribution, we construct the acquisition function $\alpha(\delta|\mathcal{D}_{t-1})$ to help select the next query point $\delta_t$. The acquisition function can be considered as a utility measure which balances the exploration and exploitation by giving higher utility to input regions where the functional value is high (high $\mu_y$) and where the model is very uncertain (high $k_y$). The approach of using BayesOpt with a GP surrogate to attack the target model is described in Algorithm 1.

---

[2]Note that our method is not dependent on a particular choice of decoder, i.e. it could also be used together with an autoencoder- or PCA-based dimensionality reduction technique.

---

**Algorithm 1** BayesOpt Attack

---

1: **Input:** A black-box function $y$, observation data $\mathcal{D}_0$, iteration budget $T$, a decoder $g(\cdot)$
2: **Output:** The best recommended adversarial example $x^* = \boldsymbol{x}_{origin} + g(\boldsymbol{\delta}^*)$
3: **for** $t = 1, \ldots, T$ **do**
4:     Select $\boldsymbol{\delta}_t = \arg\max \alpha_t(\boldsymbol{\delta}|\mathcal{D}_{t-1})$
5:     $y_t = y(\boldsymbol{\delta}_t)$ and $\mathcal{D}_t \leftarrow \mathcal{D}_{t-1} \cup (\boldsymbol{\delta}_t, y_t)$
6:     Update the surrogate model with $\mathcal{D}_t$
7: **end for**

---

### 4.3 Additive GP-based BayesOpt Attack

Although techniques such as bilinear resizing are able to reduce the input dimension from the original image size (e.g. $d = 3072$ for CIFAR10 image) to a much lower dimension (e.g. $d' = 192$), the reduced search space for the adversarial attack is still considered very high dimensional for GP-based BayesOpt (which is usually applied on problems with $d' \leq 20$).

There are two challenges in using BayesOpt for high dimensional problems. The first is the curse of dimensionality in modelling the objective function. When the unknown function is high-dimensional, estimating it with non-parametric regression becomes very difficult because it is impossible to densely fill the input space with finite number of sample points, even if the sample size is very large (Györfi et al., 2006). The second challenge is the computational difficulty in optimising the acquisition function. The computation cost needed for optimising the acquisition function to within a desired accuracy grows exponentially with dimension (Kandasamy et al., 2015).

We adopt the additive-GP model (Duvenaud et al., 2011; Kandasamy et al., 2015) to deal with the above-mentioned challenges associated with searching for adversarial perturbation in the high dimensional space. The key assumption we make is that the objective can be decomposed into a sum of low-dimensional composite functions:

$$y(\delta) = \sum_{j=1}^{M} y^{(j)}(\delta^{(A_j)}) \tag{7}$$

where $\boldsymbol{\delta}^{(A_j)}$ denotes disjoint low-dimensional subspaces, $\boldsymbol{\delta} = \cup_{j=1}^{M}\boldsymbol{\delta}^{(A_j)}$ and $\boldsymbol{\delta}^{(A_j)} \cap \boldsymbol{\delta}^{(A_i)} = \emptyset$ for all $j \neq i$. If we impose a GP prior for each $y^{(j)}$, the prior for the overall objective $y$ is also a GP: $y \sim \mathcal{GP}(\mu(\boldsymbol{\delta}), k(\boldsymbol{\delta}, \boldsymbol{\delta}'))$ where $\mu(\boldsymbol{\delta}) = \sum_{j=1}^{M} \mu^{(j)}\left(\boldsymbol{\delta}^{(A_j)}\right)$ and $k(\boldsymbol{\delta}, \boldsymbol{\delta}') = \sum_{j=1}^{M} k^{(j)}\left(\boldsymbol{\delta}^{(A_j)}, \boldsymbol{\delta}'^{(A_j)}\right)$. The predictive posterior distribution for each subspace $p(y_{1:t-1}^{(j)}|\boldsymbol{\delta}_t^{(A_j)}, \mathcal{D}_t)$ is:

$$\mu_y^{(j)}(\boldsymbol{\delta}_t^{(A_j)}) = \boldsymbol{k}^{(j)}(\boldsymbol{\delta}_t^{(A_j)}, \boldsymbol{\delta}_{1:t-1}^{(A_j)})\boldsymbol{K}_{1:t-1}^{-1}\mathbf{y}_{1:t-1},$$
$$k_y^{(j)}(\boldsymbol{\delta}_t^{(A_j)}, \boldsymbol{\delta}_t'^{(A_j)}) = k^{(j)}(\boldsymbol{\delta}_t^{(A_j)}, \boldsymbol{\delta}_t'^{(A_j)}) - \boldsymbol{k}^{(j)}(\boldsymbol{\delta}_t^{(A_j)}, \boldsymbol{\delta}_{1:t-1}^{(A_j)})\mathbf{K}_{1:t-1}^{-1}\boldsymbol{k}^{(j)}(\boldsymbol{\delta}_{1:t-1}^{(A_j)}, \boldsymbol{\delta}_t'^{(A_j)}).$$

In our case, the exact decomposition (i.e. which input dimension belongs to which low-dimensional subspace) is unknown but we can learn it together with other GP hyperparameters by maximising marginal likelihood (Kandasamy et al., 2015). We demonstrate the effectiveness of our decomposition learning in Appendix E. Note that in this case, the acquisition function is formulated based on $p(y_t^{(j)}|\boldsymbol{\delta}_t^{(A_j)}, \mathcal{D}_t)$ for each subspace and is also optimised in the low-dimensional subspace, thus leading to much more efficient optimisation task. The optimal perturbations in all the subspaces $\{\boldsymbol{\delta}^{(A_j)*}\}_{j=1}^{M}$ are then combined to give the next query point $\boldsymbol{\delta}_t$.

### 4.4 Learning The Optimal $d^r$

Generating the successful adversarial example $\boldsymbol{x} \in \mathbb{R}^d$ by searching perturbation in a reduced dimension $\boldsymbol{\delta} \in \mathbb{R}^{d^r}$ has become a popular practice that leads to significant improvement in query efficiency (Chen et al., 2017; Tu et al., 2018; Alzantot et al., 2018). However, what the optimal $d^r$ is and how to decide it efficiently have not been investigated in previous work (Chen et al., 2017; Tu et al., 2018; Alzantot et al., 2018). As we shown empirically in Section 5.1, setting $d^r$ arbitrarily can lead to suboptimal attack performance in terms of query efficiency, attack success rate as well

---

**Algorithm 2** Bayesian selection of $d^r$

---

1: **Input:** A decoder $g(\cdot)$, observation data $\mathcal{D}^d_{t-1} = \{g(\boldsymbol{\delta}_i), y_i\}_{i=1}^{t-1}$ where $g(\boldsymbol{\delta}_i) \in \mathbb{R}^d$ and a set of possible $d^r : \{d^r_j\}_{j=1}^N$
2: **Output:** The optimal reduced dimension $d^{r*}$ and the corresponding GP model
3: **for** $j = 1, \ldots, N$ **do**
4:    $\mathcal{D}^{d^r_j}_{t-1} = \{g^{-1}\left(g(\boldsymbol{\delta}_i)\right), y_i\}_{i=1}^{t-1}$ where $g^{-1}\left(g(\boldsymbol{\delta}_i)\right) \in \mathbb{R}^{d^r_j}$
5:    Fit a GP model to $\mathcal{D}^{d^r_j}_{t-1}$ and computes its maximum marginal likelihood $p(\mathcal{D}^d_{t-1}|\boldsymbol{\theta}^*, d^r_j)$
6: **end for**
7: Select $d^{r*} = \arg\max_{d^r_j \in \{d^r_j\}_{j=1}^N} p(\mathcal{D}^d_{t-1}|\boldsymbol{\theta}^*, d^r_j)$ and its correspond GP model

---

as perturbation costs while finding a good $d^r$ by trial and error or progressive increasing is computationally expensive and highly inefficient because the optimal $d^r$ varies with different attack input $\boldsymbol{x}_{origin}$. An effective way of learning $d^r$ is thus very important for adversarial attacks. In this section, we propose a rigorous method, which is neatly compatible with our attack technique, to learn the optimal $d^r$ from the query information.

The optimal $d^r$ should be the one that both takes into consideration our prior knowledge on the discrete $d^r$ choices and at the same time best explain the observed query data. Given that our BayesOpt attack uses a statistical surrogate (i.e. GP in our case) to model the unknown relation between the attack objective score $y$ and the adversarial perturbation $\boldsymbol{\delta}$, this naturally translates to the criterion of maximising the posterior for $d^r$:

$$p(d^r_i|\mathcal{D}_{t-1}) = \frac{p(\mathcal{D}_{t-1}|d^r_i)p(d^r_i)}{p(\mathcal{D}_{t-1})} \tag{8}$$

where $p(\mathcal{D}_{t-1}|d^r_j)$ is the marginal likelihood (evidence) of adopting a specific $d^r_j$, $p(d^r_j)$ is our prior over the possible $d^r$ values and $p(\mathcal{D}_{t-1}) = \sum_i p(\mathcal{D}_{t-1}|d^r_j)p(d^r_j)$ is a normalising constant. If we match different $d^r_j$ to different model choice, the problem of choosing $d^r$ then becomes the classic *Bayesian model selection* task (Rasmussen, 2003). In most cases, our prior assumption is that we do not prefer one $d^r$ over another (i.e. flat prior $p(d^r_j) = p(d^r_i)$ for $j \neq i$) and thus we can select $d^r$ by comparing their evidence(MacKay & Mac Kay, 2003):

$$\frac{p(d^r_j|\mathcal{D}_{t-1})}{p(d^r_j|\mathcal{D}_{t-1})} = \frac{p(\mathcal{D}_{t-1}|d^r_j)p(d^r_j)}{p(\mathcal{D}_{t-1}|d^r_j)p(d^r_j)} = \frac{p(\mathcal{D}_{t-1}|d^r_j)}{p(\mathcal{D}_{t-1}|d^r_j)}. \tag{9}$$

The exact computation of the evidence term requires marginalisation over model hyper-parameters, which is intractable. We approximate the integral with point estimates (i.e. marginal likelihood of our GP model): $p(\mathcal{D}_{t-1}|d^r_j) = \int p(\mathcal{D}_{t-1}|\boldsymbol{\theta}, d^r_j)p(\boldsymbol{\theta}|d^r_j)d\boldsymbol{\theta} \approx p(\mathcal{D}_{t-1}|\boldsymbol{\theta}^*, d^r_j)$ where $\boldsymbol{\theta}^* = \arg\max_{\boldsymbol{\theta}} p(\mathcal{D}_{t-1}|\boldsymbol{\theta}, d^r_j)$. For query efficiency, we project the same perturbation query data in the original image dimension $\mathcal{D}^d_{t-1} = \{g(\boldsymbol{\delta}_i), y_i\}_{i=1}^{t-1}$ to different latent spaces to get corresponding low-dimensional training data sets for separate GP models. The GP model that corresponds to $d^r_j$ is trained on $\mathcal{D}^{d^r_j}_{t-1} = \{g^{-1}\left(g(\boldsymbol{\delta}_i)\right), y_i\}_{i=1}^{t-1}$ where $g^{-1}\left(g(\boldsymbol{\delta}_i)\right) \in \mathbb{R}^{d^r_j}$. Then we select the optimal $d^r$ by comparing the marginal likelihood $p(\mathcal{D}^{d^r_j}_{t-1}|\boldsymbol{\theta}^*, d^r_j)$ of each GP surrogate. The overall procedure for $d^r$ selection is described in Algorithm 2. We would like to highlight that the use of the statistical surrogate in our BayesOpt attack approach enables us to naturally use the Bayesian model selection technique to learn the optimal $d^r$ that automatically enjoy the trade-off between data-fit quality and model complexity. And we also show empirically in Section 5.3 that by automating and incorporating the learning of $d^r$ into our BayesOpt attack, we can gain higher success rate and query efficiency. Other adversarial attacks methods can also use our proposed approach to decide $d^r$ but it would require the additional efforts of constructing statistical models to provide $p(\mathcal{D}_{t-1}|d^r_j)$.

## 5 EXPERIMENTS

We empirically compare the performance of our BayesOpt attacks against the state-of-the-art black-box methods such as ZOO (Chen et al., 2017), AutoZOOM(Tu et al., 2018) and GenAttack (Alzantot et al., 2018). We denote the BayesOpt attack with standard GP surrogate as GP-BO, its variant that

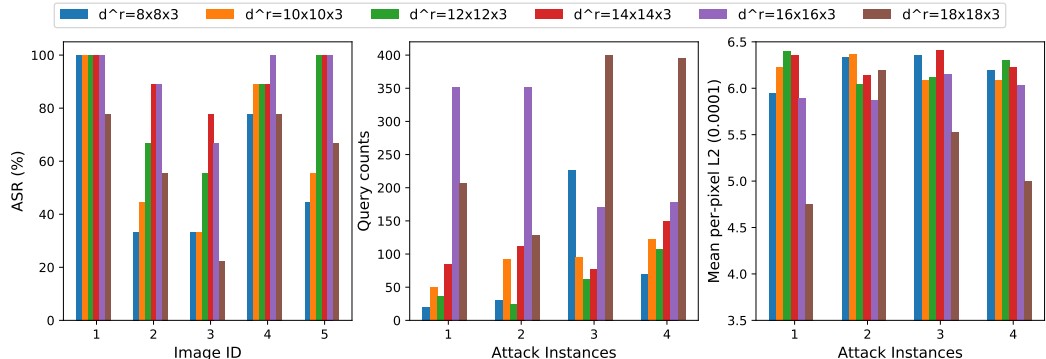

Figure 1: Performance of BayesOpt attack with GP surrogate in different reduced dimension $d^r$. Each color represents a particular $d^r$. The **left** subplot shows the attack success rates(ASR) for attacking all the other 9 classes on each of the 5 original images. The $d^r$ that leads to the highest ASR varies with the original images. The **middle** subplot shows the number of queries needed to attack the same attack instances (i.e. the same original image and the same attack target) by using different $d^r$. For the same 4 attack instances, the **right** subplot show the effect of the choice of $d^r$ on the $L_2$ distances between the resultant adversarial examples and the original image.

learns the $d^r$ automatically is GP-BO-auto-$d^r$ as well as the attack with additive GP surrogate as ADDGP-BO. For all the BayesOpt methods, we use GP-UCB as the acquisition function [3] and update the GP hyperperameters every 5 BayesOpt iterations. We relearn the optimal $d^r$ for GP-BO-auto-$d^r$ and the search space decomposition for ADDGP-BO every 40 iterations. For ADDGP-BO, we decompose the search spaces into $M = 12$ sub-spaces of equal dimensions for MNIST and CIFAR10 and into $M = 27$ sub-spaces for ImageNet.

The target models that we attack follow the same architectures as that used in AutoZOOM and GenAttack; These are image classifiers for MNIST (a CNN with 99.5% test accuracy) and CIFAR10 (a CNN with 80% test accuracy). Following the experiment design in (Tu et al., 2018), we randomly select 50 correctly classified images from CIFAR10 test data and MNIST test data. We then perform targeted attacks on these images. Each selected image is attacked 9 times, targeting at all but its true class and this gives a total of 450 attack instances for both CIFAR10 and MNIST. For ImageNet, we also select 50 correctly classified images from the test set but perform one random targeted attack for each image. We set $\delta_{max} = 0.3$ for attacking MNIST and $\delta_{max} = 0.05$ for CIFAR10 and ImageNet, which are used in (Alzantot et al., 2018). We use the recommended parameter setting and their open sourced implementations for performing all competing attack methods (Chen et al., 2017; Tu et al., 2018; Alzantot et al., 2018).

## 5.1 EFFECT OF $d^r$

We first empirically investigate the effect of the reduced dimensionality $d^r$ of the latent space in which we search for the adversarial perturbation. We experiment with the GP-based BayesOpt attacks for the CIFAR10 classifier using reduced dimension of $d_r = \{6 \times 6 \times 3, 8 \times 8 \times 3, 10 \times 10 \times 3, 12 \times 12 \times 3, 14 \times 14 \times 3, 16 \times 16 \times 3, 18 \times 18 \times 3\}$ and perform targeted attacks on 5 target images with each image being attacked 9 times, leading to 45 attack instances. We first investigate the attack success rate (ASR) achieved at different $d^r$ for all the 5 images. The results on ASR out of the 9 targeted attacks for each of the 5 images, which are indicated by Image ID 1 to 5, are shown in the left subplot of Figure 1. It's evident that the $d^r$ which leads to highest attack success rate varies for different original images $\boldsymbol{x}_{origin}$.

We then examines how the $d^r$ affects the query efficiency and attack quality (average $L_2$ distance of the adversarial image from the original image) for the attack instances (e.g. make the classifier to mis-classify a airplane image as a cat) that GP-based BayesOpt can attack successfully at all $d^r$. We present the results on 4 attack instances of a airplane image in the middle and right subplots of Figure 1. We can see that even for the same original image and attack instances, varying $d^r$ can

---

[3] The parameter that trades off exploration and exploitation is set to 2.

Table 1: Attack results on 50 random MNIST images by using $d^r = 14 \times 14 \times 1$. $Q$ denotes the query count. *ASR* denotes attack success rate. The standard errors are in parentheses.

| Attack method | ASR | Q (Max, Median, Mean) | Average $L_2$ perturbation (per pixel) |
|---|---|---|---|
| ADDGP-BO | 97% | $829, 46, 95(\pm6)$ | $6.81 \times 10^{-3}(\pm3.12 \times 10^{-5})$ |
| GP-BO-auto-$d^r$ | 98% | $833, 75, 121(\pm6)$ | $6.89 \times 10^{-3}(\pm5.55 \times 10^{-5})$ |
| GP-BO | 82% | $899, 42, 77(\pm5)$ | $7.15 \times 10^{-3}(\pm4.17 \times 10^{-5})$ |
| GenAttack | 92% | $986, 146, 217(\pm9)$ | $6.66 \times 10^{-3}(\pm2.81 \times 10^{-5})$ |
| AutoZOOM | 99% | $998, 229, 265(\pm9)$ | $5.05 \times 10^{-3}(\pm7.71 \times 10^{-5})$ |
| ZOO | 1% | $256, 128, 128(\pm64)$ | $2.78 \times 10^{-3}(\pm3.53 \times 10^{-5})$ |

Table 2: Attack results on 50 randomly selected CIFAR10 images with $d^r = 14 \times 14 \times 3$. $Q$ denotes the query count. *ASR* denotes attack success rate. The standard errors are in parentheses.

| Attack method | ASR | Q (Max, Median, Mean) | Average $L_2$ perturbation (per pixel) |
|---|---|---|---|
| ADDGP-BO | 87% | $885, 154, 222(\pm10)$ | $5.87 \times 10^{-4}(\pm3.22 \times 10^{-6})$ |
| GP-BO-auto-$d^r$ | 87% | $891, 159, 234(\pm10)$ | $5.96 \times 10^{-4}(\pm5.00 \times 10^{-6})$ |
| GP-BO | 72% | $899, 141, 190(\pm8)$ | $5.78 \times 10^{-4}(\pm4.67 \times 10^{-6})$ |
| GenAttack | 68% | $991, 246, 329(\pm14)$ | $7.38 \times 10^{-4}(\pm4.14 \times 10^{-6})$ |
| AutoZOOM | 38% | $896, 102, 154(\pm11)$ | $9.97 \times 10^{-4}(\pm1.53 \times 10^{-5})$ |
| ZOO | 4% | $768, 256, 369(\pm57)$ | $1.77 \times 10^{-4}(\pm1.16 \times 10^{-5})$ |

impact query efficiency and $L_2$ norm of the successful adversarial perturbation significantly. For example, $d^r = 8 \times 8 \times 3$ is most query efficient for attack instance 1 and 4 but is outperformed by other dimensions in attack instance 2 and 3. Therefore, the importance of $d^r$ and the difficulty of finding the optimal $d^r$ for a specific target/image motivates us to derive our method for learning it automatically from the data. As shown in the following sections, our $d^r$ learner does lead to more superior attack performance.

## 5.2 PERFORMANCE UNDER A FIXED QUERY BUDGET

In this experiment, we limit the total query number to be 1000, which is slightly above the median query counts needed for GenAttack to make a successful attack on MNIST and CIFAR10 (Alzantot et al., 2018) and 2000 for ImageNet. We adopt the reduced search dimension recommended by AutoZOOM, $d^r = 14 \times 14 \times 3$ for CIFAR10 and $d^r = 14 \times 14 \times 1$ for MNIST and allow GP-BO-auto-$d^r$ to automatically learn the reduced dimension $d^r$ in a range between $6 \times 6 \times 1$ and $28 \times 28 \times 1$ for MNIST, and between $6 \times 6 \times 3$ and $32 \times 32 \times 3$ for CIFAR10.

For ImageNet, due to the high dimensionality of its images, we adopt a hierarchical decoding process: 1) first performance BayesOpt attacks(ADDGP-BO and GP-BO) on a reduced dimension of $d_1^r = 48 \times 48 \times 3$ and then 2) decode the adversarial perturbation found in $d_1^r$ to $d_2^r = 96 \times 96 \times 3$ via bilinear upsampling. 3) This is followed by another bilinear decoder projecting the adversarial perturbation in $d_2^r$ back to image dimension of $d = 299 \times 299 \times 3$. As for the GP-BO-auto-$d^r$, we allow the algorithm to automatically learn the reduced dimension $d_1^r$ in the phase 1) in the range between $6 \times 6 \times 3$ to $60 \times 60 \times 3$. GenAttack is performed with the same upsampling as our methods. As for AutoZOOM and ZOO, we uses the optimal $d^r$ settings recommended in (Tu et al., 2018; Chen et al., 2017) for ImageNet.

For BayesOpt attack, each iteration requires 1 query to the objective function so we limit its iteration to 1000 and early terminate the BayesOpt attack algorithm when successful adversarial example is found. AutoZOOM comprises 2 stages in their attack algorithm. The first exploration stage aims to find the successful adversarial example. Once a successful attack is found, it switches to the fine-tuning stage to reduce the perturbation cost (e.g. $L_2$ norm) of the successful attack. We report its performance on the attack success rate and $L_2$ norm of adversarial perturbations after a budget of 1000 queries, which allows it to fine-tune the successful adversarial perturbations found. Moreover, AutoZOOM uses $L_2$ norm to measure the perturbation costs but our method and GenAttack limit the search space via $L_\infty$ norm. We observe that the successful attacks found by AutoZOOM incur

Table 3: Attack results on 50 randomly ImageNet images with random target label under a query budget of 2000. $Q$ denotes the query count. *ASR* denotes attack success rate. The standard errors are in parentheses. ZOO fails to make any successful attack within this budget.

| Attack method | ASR | Q (Max, Median, Mean) | Average $L_2$ perturbation (per pixel) |
|---|---|---|---|
| ADDGP-BO | 60% | $1985, 1247, 1206(\pm 90)$ | $1.74 \times 10^{-4}(\pm 1.89 \times 10^{-6})$ |
| GP-BO-auto-$d^r$ | 32% | $1999, 922, 938(\pm 128)$ | $1.96 \times 10^{-4}(\pm 8.07 \times 10^{-6})$ |
| GP-BO | 16% | $1947, 1113, 1232(\pm 162)$ | $1.62 \times 10^{-4}(\pm 7.94 \times 10^{-6})$ |
| GenAttack | 12% | $1971, 1354, 1266(\pm 259)$ | $2.03 \times 10^{-4}(\pm 9.81 \times 10^{-6})$ |
| AutoZOOM | 2% | $1451, 1451, 1451(\pm 0.00)$ | $1.21 \times 10^{-5}(\pm 0.00)$ |

much higher perturbation costs in terms of $L_\infty$ norm than GenAttack and our method. Therefore, we only consider the final adversarial examples whose $L_\infty$ distances from the original images lie within $[-\delta_{max}, \delta_{max}]$ as the successful attacks [4].

The results on MNIST in Table 1 show that all our BayesOpt attacks can achieve a comparable success rate but at a much lower query count in comparison to GenAttack (Alzantot et al., 2018) and AutoZOOM (Tu et al., 2018). Specifically, the median query count of ADDGP-BO is 68% less than GenAttack and 80% less than AutoZOOM while that of GP-BO-auto-$d^r$ is 49% less than GenAttack and 67% less than AutoZOOM.

As for results on attacking on CIFAR10, Table 2 shows that all our BayesOpt attack can achieve significantly higher success rate but again at a remarkably lower query counts than the existing black-box approaches. For example, our ADDGP-BO can achieve 18% higher success rate while using 37.4% less queries in terms of the median as compared to GenAttack. In addition, our approaches also lead to better quality adversarial examples (Figure 3 in Appendix B ) which are closer to the original image than the benchmark methods as reflected by the lower average $L_2$ perturbation (20.5% less). More importantly, this set of experiments also demonstrate the effectiveness of our Bayesian method for learning $d^r$ as GP-BO-auto-$d^r$ leads to 15% increase in attack success rate compared to GP-BO while maintaining the competitiveness in query efficiency and $L_2$ distance.

Similarly, the results on ImageNet in Table 3 shows that all our BayesOpt attacks can achieve higher attack success rate than the competing methods given a query budget of 2000. ADDGP-BO is the clearly best performing method in this very high-dimensional setting, achieving 5 times higher success rate than the best competing method, GenAttack, while again obtaining successful adversarial perturbations with lower average $L_2$ perturbation (14 % less) than GenAttack. This confirms our hypothesis that the additive-GP surrogate together with decomposition learning is very effective for high-dimensional optimisation.

### 5.3 QUERY EFFICIENCY COMPARISON

We finish by comparing the query efficiency, measuring the change in the attack success rate over query counts for all the methods. We limit the query budget of our BayesOpt attacks to 1000 but let the competing methods to continue running until they achieve the same best attack success rate as our best BayesOpt attacks or exceeds a much higher limit (2000 queries for MNIST and 5000 queries for CIFAR10 and 14000 queries for ImageNet).

As shown in Figure 2, BayesOpt attacks converge much faster to the high attack success rates than the other methods. Specifically, GP-BO-auto-$d^r$ take less than 833 queries to achieve the success rate of 98% for MNIST, which is 24% of that by GenAttack (2500). As for CIFAR10, both ADDGP-BO and GP-BO-auto-$d^r$ takes around 890 queries to achieve a success rate of 87% which is 23% of the query count by AutoZOOM and 17% of that by GenAttack. One point to note is that AutoZOOM appears to be slightly more query efficient than GenAttack in this set of experiments. However, we need to bear in mind that although we limit the mean $L_{inf}$ norm of the adversarial perturbation found by AutoZOOM to $\delta_{max}$, AutoZOOM still have the advantage of exploring beyond the $\delta_{max}$ for many dimensions. In the case of ImageNet, all our BayesOpt attacks again enjoy faster convergence, with

---

[4]This still gives an advantage to ZOO and AutoZOOM because it allows them to inject perturbation whose magnitude is larger than $\boldsymbol{\delta}_{max}$ at certain pixels but our BayesOpt methods and GenAttack limit perturbation at all the pixels to be less than $\boldsymbol{\delta}_{max}$.

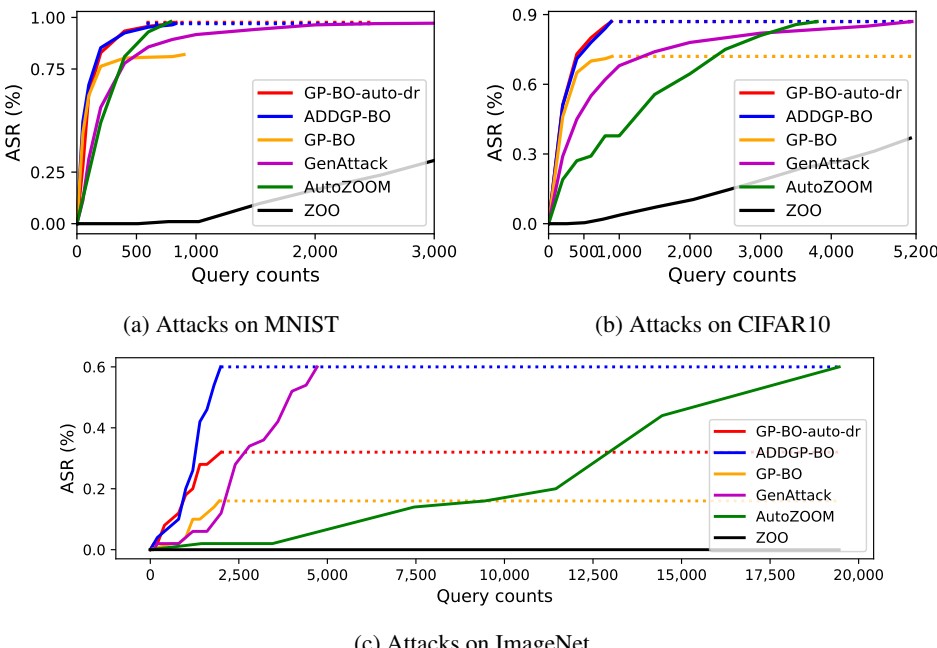

(a) Attacks on MNIST

(b) Attacks on CIFAR10

(c) Attacks on ImageNet

Figure 2: Query efficiency of BayesOpt Attacks. The plots show the attack success rate (ASR) of different methods up to certain query counts. The best BayesOpt attacks (i.e. GP-BO-auto-$d^r$ (red) and ADDGP-BO (blue)) can achieve an ASR of 98% on MNIST(a) and 87% on CIFAR10 (b) within a budget of 1000 queries. To achieve the same success rates, GenAttack (purple) takes 3500 for MNIST and 5141 for CIFAR10, and AutoZOOM (green) takes 800 for MNIST and 3880 for CIFAR10. For ImageNet (c), the best BayesOpt attack is ADDGP-BO (blue), which achieves an ASR of 60% with 1985 queries. To achieve the same success rates, GenAttack (purple) takes a budget of 4711 queries and AutoZOOM (green) takes 19451 queries. ZOO fails to make any attack on ImageNet within the given budget.

ADDGP-BO taking only 1985 queries to achieve 60% ASR. Meanwhile, GenAttack, takes a budget of 4711 queries, 2.4 times that of ADDGP-BO, to achieve the same ASR and AutoZOOM costs 19451 queries which is 9.8 times that of ADDGP-BO.

# 6 CONCLUSION

We introduce a new black-box adversarial attack which leverages Bayesian optimisation to find successful adversarial perturbations with high query efficiency. We also improve our attack by adopting an additive surrogate structure to ease the optimisation challenge over the typically high-dimensional task. Moreover, we take full advantage of our statistical surrogate model and the available query data to learn the optimal degree of dimension reduction for the search space via Bayesian model selection. In comparison to several existing black-box attack methods, our BayesOpt attacks can achieve high success rates with 2-5 times fewer queries while still producing adversarial examples that are closer to the original image (in terms of average $L_2$ distance). We believe our BayesOpt attacks can be a competitive alternative for accessing the model robustness.

One limitation of our attack methods is that due to the poor scalability of GPs, the attack algorithms are computationally more expensive than most existing alternatives especially as the query number increases. Thus, our BayesOpt attacks are optimal for the setting where the cost of evaluating the target model, being it the monetary costs, computational costs or the risk of being detected, is much higher than the computational cost of the attack algorithm itself and thus query efficiency is highly prioritised. On the other hand, replacing the GP surrogates with more scalable models such as sparse GPs or Bayesian neural networks can be an interesting future research direction.

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

## A    BAYESIAN OPTIMISATION

Here we present a generic algorithm for Bayesian optimisation.

---
**Algorithm 3** BayesOpt Algorithm
---
1: **Input:** A black-box function $y$, observation data $\mathcal{D}_0$, iteration budget $T$
2: **Output:** The best recommendation $\delta^*$
3: **for** $t = 1, \ldots, T$ **do**
4:     Select $\delta_t = \arg\max \alpha_t(\delta | \mathcal{D}_{t-1})$
5:     $y_t = y(\delta_t)$ and $\mathcal{D}_t \leftarrow \mathcal{D}_{t-1} \cup (\delta_t, y_t)$
6:     Update the surrogate model with $\mathcal{D}_t$
7: **end for**

---

## B    ADVERSARIAL EXAMPLES FOR CIFAR10

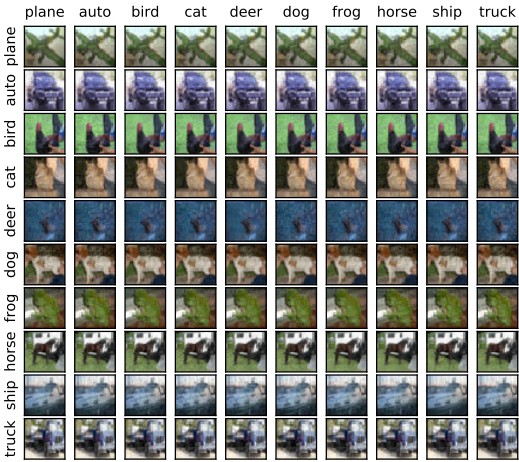

Figure 3: CIFAR10 adversarial examples generated by our BayesOpt attack. True labels and target labels correspond to the rows and columns.

## C    ADDITIONAL EXPERIMENTS ON MNIST AND CIFAR10

## D    OBJECTIVE VALUE OVER BAYESOPT ITERATIONS

We illustrate this via the case of attacking a CIFAR10 image of label 9(class truck) on the other 9 target labels with original label 9We plot the value of objective function (Equation 2), which is equal to the negative of attack loss, against the BayesOpt iterations/query counts. We can see that our additive GP surrogate (ADDGP-BO) as well as the Bayesian learning of optimal $d^r$ (GP-BO-auto-$d^r$) lead to faster convergence and thus higher attack success rate for this instance.

## E    EFFECTIVENESS OF DECOMPOSITION LEARNING

To learn the decomposition for the additive-GP surrogate in ADDGP-BO attack, we follow the approach proposed in (Kandasamy et al., 2015) to treat the decomposition as an additional hyperparameter and learn the optimal decomposition by maximising marginal likelihood. However, exhaustive search over all $M!d!/(d_s!^M)^5$ possible decompositions is expensive. We adopt a computationally cheap alternative by randomly selecting 20 decompositions and choosing the one with

---
[5] $M$ is the number of subspaces and $d_s = |A_j|$ is the dimension of each subspaces

Table 4: Summary of attack results on 7 MNIST images. $Q$ denotes the query count. *ASR* denotes attack success rate. The standard errors are in parentheses.

| Attack method | $d_r$ | *ASR* | *Q (Max, Median, Mean)* | Average $L_2$ perturbation (per pixel) |
|---|---|---|---|---|
| GP-BO | $14 \times 14 \times 1$ | 92% | $899, 53, 119(\pm28)$ | $7.01 \times 10^{-3}(\pm1.13 \times 10^{-4})$ |
| ADDGP-BO | $14 \times 14 \times 1$ | 98% | $584, 34, 67(\pm13)$ | $6.50 \times 10^{-3}(\pm8.39 \times 10^{-5})$ |
| AutoZOOM | $14 \times 14 \times 1$ | 97% | $892, 186, 247(\pm27)$ | $5.10 \times 10^{-3}(\pm1.92 \times 10^{-4})$ |
| GenAttack | $14 \times 14 \times 1$ | 97% | $976, 184, 239(\pm22)$ | $5.56 \times 10^{-3}(\pm9.25 \times 10^{-5})$ |
| GP-BO | $16 \times 16 \times 1$ | 97% | $400, 50, 80(\pm12)$ | $7.08 \times 10^{-3}(\pm1.09 \times 10^{-4})$ |
| ADDGP-BO | $16 \times 16 \times 1$ | 100% | $540, 42, 72(\pm12)$ | $6.49 \times 10^{-3}(\pm9.41 \times 10^{-5})$ |
| AutoZOOM | $16 \times 16 \times 1$ | 98% | $812, 251, 295(\pm28)$ | $5.98 \times 10^{-3}(\pm2.03 \times 10^{-4})$ |
| GenAttack | $16 \times 16 \times 1$ | 97% | $928, 202, 237(\pm19)$ | $5.61 \times 10^{-3}(\pm8.58 \times 10^{-5})$ |
| GP-BO | $28 \times 28 \times 1$ | 98% | $430, 201, 225(\pm11)$ | $8.71 \times 10^{-3}(\pm1.94 \times 10^{-4})$ |
| ADDGP-BO | $28 \times 28 \times 1$ | 100% | $666, 57, 100(\pm15)$ | $9.71 \times 10^{-3}(\pm1.12 \times 10^{-4})$ |
| AutoZOOM | $28 \times 28 \times 1$ | 98% | $860, 185, 244(\pm27)$ | $1.01 \times 10^{-2}(\pm1.63 \times 10^{-4})$ |
| GenAttack | $28 \times 28 \times 1$ | 83% | $976, 365, 399(\pm33)$ | $6.26 \times 10^{-3}(\pm6.69 \times 10^{-5})$ |

Table 5: Summary of attack results on 27 CIFAR10 images. $Q$ denotes the query count. *ASR* denotes attack success rate. The standard errors are in parentheses.

| Attack method | $d_r$ | *ASR* | *Q (Max, Median, Mean)* | Average $L_2$ perturbation (per pixel) |
|---|---|---|---|---|
| GP-BO | $8 \times 8 \times 3$ | 52% | $314, 48, 70(\pm13)$ | $5.78 \times 10^{-4}(\pm1.44 \times 10^{-5})$ |
| ADDGP-BO | $8 \times 8 \times 3$ | 75% | $899, 140, 234(\pm33)$ | $5.55 \times 10^{-4}(\pm8.46 \times 10^{-6})$ |
| AutoZOOM | $8 \times 8 \times 3$ | 56% | $382, 126, 139(\pm17)$ | $9.58 \times 10^{-4}(\pm2.66 \times 10^{-5})$ |
| GenAttack | $8 \times 8 \times 3$ | 67% | $671, 236, 286(\pm32)$ | $7.40 \times 10^{-4}(\pm1.30 \times 10^{-5})$ |
| GP-BO | $14 \times 14 \times 3$ | 81% | $899, 142, 192(\pm11)$ | $5.74 \times 10^{-4}(\pm6.96 \times 10^{-6})$ |
| ADDGP-BO | $14 \times 14 \times 3$ | 90% | $885, 141, 213(\pm15)$ | $5.79 \times 10^{-4}(\pm5.16 \times 10^{-6})$ |
| AutoZOOM | $14 \times 14 \times 3$ | 42% | $376, 95, 133(\pm12)$ | $9.82 \times 10^{-4}(\pm1.95 \times 10^{-5})$ |
| GenAttack | $14 \times 14 \times 3$ | 75% | $971, 251, 321(\pm21)$ | $7.35 \times 10^{-4}(\pm6.92 \times 10^{-6})$ |
| GP-BO | $16 \times 16 \times 3$ | 83% | $785, 280, 292(\pm22)$ | $5.34 \times 10^{-4}(\pm1.43 \times 10^{-5})$ |
| ADDGP-BO | $16 \times 16 \times 3$ | 87% | $878, 168, 229(\pm29)$ | $5.76 \times 10^{-4}(\pm8.93 \times 10^{-6})$ |
| AutoZOOM | $16 \times 16 \times 3$ | 3% | $298, 170, 170(\pm90)$ | $7.50 \times 10^{-4}(\pm1.35 \times 10^{-5})$ |
| GenAttack | $16 \times 16 \times 3$ | 79% | $986, 281, 339(\pm36)$ | $7.36 \times 10^{-4}(\pm1.55 \times 10^{-5})$ |

the largest marginal likelihood. The method decomposition learning procedure is repeated every 40 BO iterations and we use $M = 12$ for CIFAR10. We denote the method as ADDGP-BO-LD in this section but as ADDGP-BO in the rest of the paper.

We also experiment with another alternative way to learn the decomposition (ADDGP-BO-FD), which is similar to importance sampling in ZOO. We group the pixels/dimensions together if the magnitude of average change in their pixel values over the past 5 BO iterations are closer (i.e. pixels that are subject to the most large adversarial perturbations are grouped together). Again, we divide the dimensions into disjoint 12 groups as above to ensure fair comparison.

Table 6: Summary of attack results on CIFAR10 for different decomposition learning method. $d_r = 14 \times 14 \times 3$ which is further decomposed into 12 subspaces of $d_s = 49$. $Q$ denotes the query count. *ASR* denotes attack success rate. The standard errors are in parentheses.

| Attack method | *ASR* | *Q (Max, Median, Mean)* | Average $L_2$ perturbation (per pixel) |
|---|---|---|---|
| ADDGP-BO-LD | 94% | $885, 134, 212(\pm16)$ | $5.78 \times 10^{-4}(\pm5.32 \times 10^{-6})$ |
| ADDGP-BO-FD | 87% | $899, 143, 196(\pm14)$ | $5.05 \times 10^{-4}(\pm4.52 \times 10^{-6})$ |

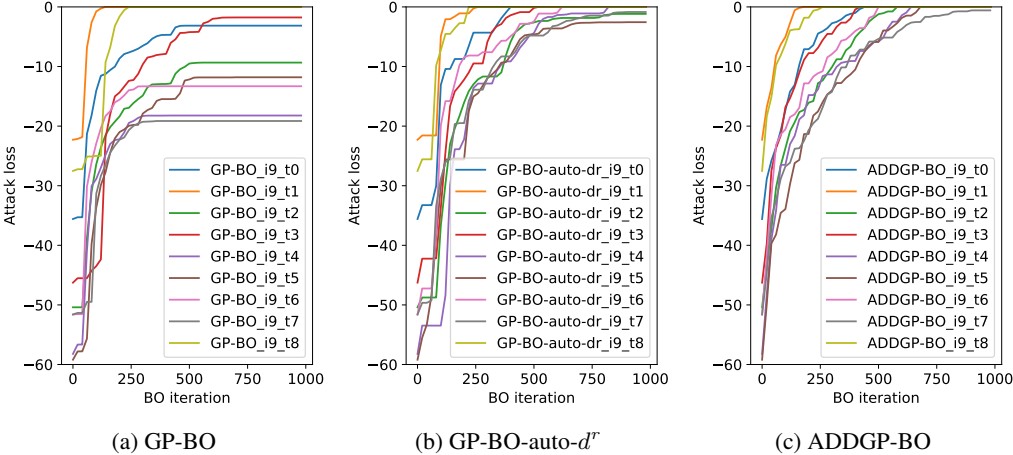

(a) GP-BO            (b) GP-BO-auto-$d^r$           (c) ADDGP-BO

Figure 4: Attack objective value against BayesOpt iterations(query count) for using various BayesOpt methods to attack one CIFAR10 image of class label 9 (denoted by i). Curves of different colours correspond to the 9 different target labels (denoted by t). Convergence to 0 objective value indicates successful attack. ADDGP-BO and GP-BO-auto-$d^r$ enjoy faster convergence and thus higher attack success rates than simple GP-BO.

We compare the both types of decomposition learning methods using 20 randomly selected CIFAR10 images, each of which is attacked on 9 other classes except its original class and a query budget of 1000. The results are shown in Table 6. It is evident that learning decomposition by maximising marginal likelihood (ADDGP-BO-LD) can achieve higher attack success rate than pixel-value-change-based decomposition learning (ADDGP-BO-FD) given the query budget.

## F  GAUSSIAN PROCESSES

Gaussian processes (GPs) are popular models for inference in regression problems, especially when a quantification of the uncertainty around predictions is of relevance and little prior information is available to the inference problem. These qualities, together with their analytical tractability, make them the most commonly used surrogate model in Bayesian optimisation. A comprehensive overview on GPs can be found in Rasmussen & Williams (2006). Below we highlight the concepts of marginal likelihood and how to use it for hyperparameter optimisation and model selection based on the notation introduced in Section 4.2.

**Hyperparameter tuning.**  The key to finding the right hyperparameters $\boldsymbol{\theta}^{*}$ in light of $\mathcal{D}_{t-1}$ in a principled way is given by the marginal likelihood in Equation (10).

$$p(\mathcal{D}_{t-1}|\boldsymbol{\theta}) = (2\pi)^{-\frac{t-1}{2}}|\mathbf{K}_{1:t-1}(\boldsymbol{\theta})|^{-\frac{1}{2}} \exp\left(-\frac{1}{2}\boldsymbol{y}_{1:t-1}^{T}\mathbf{K}_{1:t-1}^{-1}(\boldsymbol{\theta})\boldsymbol{y}_{1:t-1}\right), \qquad (10)$$

where we have introduced a slightly augmented notation of $\mathbf{K}_{1:t-1} \equiv \mathbf{K}_{1:t-1}(\boldsymbol{\theta})$ to highlight the dependence on the kernel hyperparamaters $\boldsymbol{\theta}$. In a truly Bayesian approach, one could shy away from fixing one set of hyperparameters and use Equation (10) to derive a posterior distribution of $\boldsymbol{\theta}$ based on one's prior beliefs about $\boldsymbol{\theta}$ expressed through some distribution $p_0(\boldsymbol{\theta})$. However, as such an approach generally requires sampling using methods like Markov chain Monte Carlo (MCMC) due to intractability of integrals, in practice it is often easier and computationally cheaper to replace the fully Bayesian approach by a maximum likelihood approach and simply maximise the likelihood $p(\mathcal{D}_{t-1}|\boldsymbol{\theta})$ w.r.t. $\boldsymbol{\theta}$. We follow this approach and perform a mix of grid- and gradient based search to maximise the logarithm of the r.h.s. of Equation (10). After evaluating a grid of $5'000$ points, we start gradient ascent on each of the 5 most promising points using the following equation for the gradient (Rasmussen & Williams, 2006) to find the hyperparameters $\boldsymbol{\theta}^{*}$ which maximise the marginal likelihood:

$$\frac{\partial}{\partial \theta_j} \log p(\mathcal{D}_{t-1}|\boldsymbol{\theta}) = \frac{1}{2}\boldsymbol{y}_{1:t-1}^T \mathbf{K}_{1:t-1}^{-1} \frac{\partial \mathbf{K}_{1:t-1}}{\partial \theta_j} \mathbf{K}_{1:t-1}^{-1} \boldsymbol{y}_{1:t-1} - \frac{1}{2} tr\left(\mathbf{K}_{1:t-1}^{-1} \frac{\partial \mathbf{K}_{1:t-1}}{\partial \theta_j}\right) \quad (11)$$

$$= \frac{1}{2} tr\left((\mathbf{K}_{1:t-1}^{-1} \boldsymbol{y}_{1:t-1} \boldsymbol{y}_{1:t-1}^T \mathbf{K}_{1:t-1}^{-1} - \mathbf{K}_{1:t-1}^{-1}) \frac{\partial \mathbf{K}_{1:t-1}}{\partial \theta_j}\right) \quad (12)$$

**Model selection.** Once the optimal hyperparameters $\boldsymbol{\theta}^*$ are found as described above, they can be plugged back into Equation (10) to choose amongst different models. In Section 4.4 this is described for the case of choosing between different values of the reduced dimensionality $d^r$.

