# OpenReview forum: "BayesOpt Adversarial Attack"
_ICLR.cc/2020/Conference — Accept (Poster)_

### Official Review · AnonReviewer3 · 2019-10-22
**Official Blind Review #3**

**Rating:** 3

**Review:**

In this paper, the authors propose to use Bayesian optimization with a GP surrogate for adversarial image generation. In addition to the standard BayesOpt algorithm, the authors use a variant that exploits additive structure, as well as a variant that uses Bayesian model selection to determine an optimal dimensionality reduction.

For the experimental results, I find it extremely surprising that vanilla GP-BO works at all, even downsampling e.g. to d=588 (Table 2). This is extraordinarily high dimensionality for vanilla BayesOpt, and conventional wisdom suggests that this should not work at all. I'd like to see a discussion of this, particularly as I've seen unsuccessful attempts at this in the past. What differentiating factors lead to it working here? The set of images considered is quite small, presumably because of the rather extreme wall clock expense of running hundreds of sequential BayesOpt iterations without GPU acceleration. This is particularly true for methods that require Bayesian model selection and therefore training multiple GPs in each iteration of BayesOpt.

Along the same lines of dimensionality concerns, I would view a lack of results on ImageNet images as a significant weakness, particularly as these are probably much harder for general purpose blackbox optimizers, as the initial dimensionality of those images is ~150000. A decent amount of missing related literature studies transformations of ImageNet images, including the QL Attack (Ilyas et al., 2018), Bandits-TD (Ilyas et al., 2019) and others. These papers also focus specifically on query budget, so it would be hard to claim that BayesOpt is SOTA if it can't scale to images this large.

Can you provide additional details on the learning mechanism for the additive decomposition? Are you learning kernel outputscales for different predefined additive components as in Duvenaud et al., 2011? Note that this is a fairly different structure than considered in Kandasamy et al., 2015 (despite both being called "additive GPs") -- the type of additive structure in Kandasamy et al., 2015 usually needs to be learned through approximate model selection mechanisms (usually via Metropolis-Hastings or Gibbs sampling).

**Experience Assessment:**

I have published in this field for several years.

**Review Assessment: Checking Correctness Of Derivations And Theory:**

I carefully checked the derivations and theory.

**Review Assessment: Checking Correctness Of Experiments:**

I carefully checked the experiments.

**Review Assessment: Thoroughness In Paper Reading:**

I read the paper thoroughly.

---

> ### Author Response · Authors · 2019-11-12
> **Reply to Reviewer #3**
>
> We thank the reviewer for his insightful comments. We address the concerns below:
>
> 1. "Need a discussion on the surprising phenomenon that vanilla GP-BO can work at all for such problem of extraordinarily high dimensionality."
> Please refer to our reply to all Reviewers for the detailed discussion.
>
> 2. "a lack of results on ImageNet images, which is much harder for general purpose blackbox optimizers given the initial dimensionality of those images is ~150000"
> To verify the feasibility/applicability of using our BayesOpt methods to perform "targeted" attacks on ImageNet, we select 50 correctly classified images from the ImageNet test data and perform random targeted attacks with a query budget of 2000. We found that direct application of the BayesOpt attacks on the ImageNet image to do targeted attack rarely work due to the extremely high dimensionality of search space. However, we experimented a hierarchical decoding process:
>   a) first performance BayesOpt (ADDGP-BO) on a reduced dimension of  d^r_1=48x48x3 or perform GP-BO-auto-dr by learning the optimal reduced dimension in the range up to d^r_1=48x48x3 and then
>   2) decode the adversarial perturbation found in d^r_1 to d^r_2=96x96x3 via bilinear upsampling.
>   3) This is followed by another bilinear decoder projecting the adversarial perturbation in d^r_2 back to image dimension of d=299x299x3.
> Such hierarchical decoding leads to an ASR of 60% by ADD-GP-BO and an ASR of 32% by GP-BO-auto-dr, which are higher than the ASR of 12% by GenAttack on the same image-target pairs with the same upsampling.
>
> We conduct further experiments on for our ADDGP-BO and GenAttack. ADDGP-BO achieves 60% ASR within 1985 queries but GenAttack takes 4711 (2.4 times more) queries to achieve the same ASR (See Section E in the Appendix). We will update the paper with more experimental results on ImageNet.
>
> In addition, another ICLR 2020 submission titled "Black-box Adversarial Attacks with Bayesian Optimization" has empirically demonstrated the superior query efficiency of vanilla GP-BO on ImageNet dataset in the "untargeted" attack setting.
>
> 3. "… missing related literature studies including the QL Attack (Ilyas et al., 2018), Bandits-TD (Ilyas et al., 2019)..."
> We thank the reviewer for the additional references. Bandits-TD (Ilyas et al., 2019) focuses on the simpler case of untargeted attacks and another ICLR 2020 submission titled Black-box Adversarial Attacks with Bayesian Optimization has demonstrated on ImageNet dataset that their simple GP-based BO attack, together with upsampling, (which is almost the same as the GP-BO baseline in our paper) can achieve higher attack success rate than Bandits-TD and Parsimonious attack (Moon et al., 2019) under a small query budget of 200 for the "untargeted" attack setting. And Du et al., (2019) (https://arxiv.org/abs/1906.02398) has shown that our baseline method, AutoZOOM, is more query efficient than Bandits-TD for MNIST, CIFAR10 and tiny ImageNet.  We didn’t compare against QL Attack (Ilyas et al., 2018) because two of our baseline methods, GenAttack and AutoZOOM, had shown to be more query efficient than QL Attack in (Alzantot et al., 2018). In addition, both QL Attack and Bandits-TD require gradient estimation while our proposed method doesn’t.
>
> 4. "… additional details on learning the decomposition for the additive GP surrogate.... the additive structure in Kandasamy et al., 2015 usually needs to be learned through Metropolis-Hastings or Gibbs sampling."
> We follow the approach proposed in (Kandasamy et al., 2015) to treat the decomposition as an additional hyperparameter and learn the optimal decomposition by maximising marginal likelihood. However, exhaustive search over all possible (M!d!/(d_s!^M)) decompositions (i.e. decomposing d-dimensional space into M subspaces of d_s dimensions) is expensive. We adopt a computationally cheap alternative by randomly selecting 20 decompositions and choosing the one with the largest marginal likelihood. The decomposition learning procedure is repeated every 40 BO iterations. As the reviewer mentioned, sophisticated sampling procedures can also be used to learn the decomposition and usually lead to better performance. However, they are computationally much more expensive than the maximum marginal likelihood approach.
>
> We verified the effectiveness of our way of learning decomposition by testing another alternative way to learn the decomposition; pixels are grouped together if the magnitude of change in their pixel values over iterations are close. This is similar to importance sampling in ZOO. The performance of such pixel-value-change-based decomposition learning gives lower attack success rate than our approach of learning the decomposition via marginal likelihood. We have added this comparison as Section D in the Appendix.

---

### Official Review · AnonReviewer1 · 2019-10-23
**Official Blind Review #1**

**Rating:** 6

**Review:**

This paper studied the problem of black-box adversarial attack generation by leveraging Bayesian optimization (BO).

Merits of this paper:
1) The combination of BO and dimension reduction, which makes BO more efficient under a low-dimensional space.
2) Good experiment results.

Comments/questions about this paper:

1) Comment on "Finally, to the best of our knowledge, the only prior work that uses Bayesian optimisation is a workshop paper by...". BO was also used for generating black-box adversarial examples at https://arxiv.org/pdf/1907.11684.pdf
This is a missing related work, and please elaborate on the differences.

2) The presentation of the proposed algorithm is not clear. Please explicitly state the acquisition function. And how to tune the hyperparameter in the acquisition function? What decoder is used in experiments? Have authors tested the sensitivity of the decoder (not reduced dimension)?

3) In experiments, the authors mentioned "we randomly select 3 correctly classified images for each class from CIFAR10 test data which sums up to 27 CIFAR10 images, and randomly select 7 correctly classified images from MNIST test data."
I feel that the number of tested images is not sufficient. How about conducting experiments on a large number of tested images for untargeted attack?

4) In Table 1, what does 0,0,0, mean in ZOO?

5) It is known that BO has itself parameter to tune, and is not computationally efficient. It might be good to show the computation efficiency of BO for different reduced dimensions together with the corresponding attack performance.

6) The convergence of BO is usually not stable. However, Figure 3 shows that BO converges very smoothly in terms of ASR. Could authors also show the loss value of using BO-attacks against iteration numbers?
Meanwhile, in Figure 3 is the best ASR  (up to the current query counts) reported or the ASR at the current query number?


Based on the aforementioned questions, my initial rating is weak reject.


############## Post-feedback ################
Thanks for the response. Most of my questions have been addressed. Thus, I increased my score to 6.
I suggested to have a clearer presentation on the possible pros and cons of BO in attack generation, e.g., making a comparison between BO and other methods in both query efficiency and computation efficiency.


**Experience Assessment:**

I have published in this field for several years.

**Review Assessment: Checking Correctness Of Derivations And Theory:**

I assessed the sensibility of the derivations and theory.

**Review Assessment: Checking Correctness Of Experiments:**

I assessed the sensibility of the experiments.

**Review Assessment: Thoroughness In Paper Reading:**

I read the paper at least twice and used my best judgement in assessing the paper.

---

> ### Author Response · Authors · 2019-11-12
> **Reply to Reviewer #1**
>
> We thank the reviewer for his helpful comments. We address the concerns below:
>
> 1. “Difference to https://arxiv.org/pdf/1907.11684.pdf"
> We thank the reviewer for pointing out this additional reference (Zhao et al., 2019). We have cited it and highlighted the differences in Section 2. The method proposed in that work, BO-ADMM, is effectively similar to our vanilla GP-BO but without the use of decoder to reduce the search space dimension. We propose improvements tailored to the commonly high-dimensional nature of adversarial attack problems, thus further enhancing the efficiency of BayesOpt attack for such application. Specifically, the key differences between our work and their work are:
>     a) BO-ADMM applies GP-based BO directly on the space of image dimension to minimise the joint objective of attack loss and distortion loss. This makes the problem much harder for BO and leads to low-quality adversarial examples (mean L_{\inf}=0.62 for CIFAR10).  Our BayesOpt attacks minimise the attack loss under the L_{\inf}-norm constraint, and uses a decoder to reduce the search to a low-dimensional latent space, making the problem more amenable to GP-based BO. As a result, even our vanilla GP-BO, can find the adversarial examples with much smaller distortion (mean L_{\inf} = 0.028 for CIFAR10).
>     b) The method proposed in their work, BO-ADMM, only uses the simple Gaussian process as the BO surrogate. However, we propose a method to explicitly handle the high-dimensional search space effectively and build query efficient attacks by using an additive GP surrogate. This allows us to further decompose the reduced latent search space into low-dimensional subspaces. As shown in Table 2 of Section 5.2, our ADDGP-BO attack achieves much higher success rate (14% higher) given the same query budget compared to previous data efficient approaches (GP-BO).
>     c) We further propose the use of a Bayesian model selection method to learn the optimal dimensionality of the latent space in the process of optimisation/on the fly. Such Bayesian learning of the reduced dimensionality integrates naturally with a BayesOpt attack by employing the statistical surrogate but can also be applied independently with other adversarial attack methods to decide the reduced dimensionality in a principled way. We further demonstrate its effectiveness in our experiments. As shown in Table 2 of Section 5.2, it leads to 15% increase in ASR for GP-BO.
>
> 2. “What’s the acquisition function and its hyperparameter? What decoder is used? The importance of the decoder?”
> We use the UCB and set the exploration-exploitation parameter to be a constant of 2, which is adopted in BO packages such as GPyOpt. We briefly explored the use of different acquisition functions such as EI and found similar performance. We have added these clarifications in our paper.
>
> We adopt bilinear interpolation as the decoder. Please refer to Response 3 to Reviewer #2 for more details.
>
> The use of the decoder is essential for the query efficiency of BayesOpt attacks because it helps reduce the BO search space significantly. For example, as shown in Table 3 in Section B of the Appendix for MNIST, the use of decoder reduces the median query count for GP-BO from 201 (d=28x28x1) to 53 (d^r =14x14x1) to achieve comparable attack success rate.
>
> 3. “the number of tested images is not sufficient”
> We conducted more experiments on 50 random CIFAR-10 images. Please refer Response 2 to Reviewer #2.
>
> 4. “In Table 1, what does 0,0,0, mean in ZOO?”
> Sorry for the confusion. This means ZOO succeeds in attacking the 2 simplest image-target pairs at its first batch (batch size of 128) of adversarial perturbations but fails to make successful attacks on the other cases under the budget constraints.
>
> 5. “BO is not computationally efficient”
> We update the GP hyperparameters every 5 iterations and relearn the reduced dimension or the additive decomposition every 40 iterations to reduce the computational cost. BO algorithms are indeed computationally more expensive than most adversarial methods. That’s why, as highlighted in the introduction, we focus on the adversarial setting where the cost of evaluating the target model, being it the monetary costs, computational costs or the risk of being detected, is much higher than the computational cost of BO algorithm itself and thus query efficiency is highly prioritised.
>
> 6. “plot the attack loss value vs BO iterations... Clarification on Figure 3.”
> We have added the plots on the objective value (the negative of loss) against BayesOpt iterations(query count) for various BayesOpt methods in Section C of the Appendix. They show the case of attacking a CIFAR10 image of label class 9 and how our proposed modifications (additive GP and Bayesian learning of d^r) can lead to better convergence.  Figure 3 in the paper shows the ASR up to the current query counts. We have clarified this in the Figure caption.

---

### Official Review · AnonReviewer2 · 2019-10-27
**Official Blind Review #2**

**Rating:** 6

**Review:**

The paper proposes a black-box attack method that optimises both the adversarial perturbation and the optimal dimensionaity reduction in a Bayesian Optimization framework. The formulation seem sound and the experiments show improvements wrt competitors in terms of performance and query efficiency with comparable attack success rates.

* In section 4.3 the authors claim that the additive surrogate makes the GP-based BO able to deal with the problem of high dimensionality. Given that the typical dimensionality for BayesOpt is d <= 20, how are the experiments with dimensions up to 14x14x3 provided for GP-BO and GP-BO-auto-dT performed?

* The image selection protocol seems arbitrary and it does not correspond to the Tu et al. protocol which selects 50 random images from  CIFAR-100 and MNIST.

* I feel the experiments lack some details: which is the decoder used for dimensionality reduction?



**Experience Assessment:**

I have read many papers in this area.

**Review Assessment: Checking Correctness Of Derivations And Theory:**

I did not assess the derivations or theory.

**Review Assessment: Checking Correctness Of Experiments:**

I assessed the sensibility of the experiments.

**Review Assessment: Thoroughness In Paper Reading:**

I read the paper at least twice and used my best judgement in assessing the paper.

---

> ### Author Response · Authors · 2019-11-11
> **Reply to Reviewer #2**
>
> We thank the reviewer for the positive feedback and would like to address the issues raised.
>
> 1. "Given that the typical dimensionality for BayesOpt is d <= 20, how are the experiments with dimensions up to 14x14x3 provided for GP-BO and GP-BO-auto-dr performed?"
> We use a GP kernel without ARD and learn the GP hyperparameters every 5 BO iterations. The optimal reduced dimension d^r is updated every 40 iterations. Please refer to our reply to all Reviewers for a more detailed discussion.
>
> 2. "The image selection protocol does not correspond to the Tu et al. protocol which selects 50 random images from CIFAR-10."
> We conducted more experiments by selecting 50 random images from CIFAR-10 and attacking each image on the other 9 classes except its original class. We have updated the Table 2 and Figure 3 in Section 5.2 with new CIFAR10 results. Note that the relative ranking among different methods is the same as the original results in the paper and the magnitude of improvement in query efficiency and L_2 norm by BayesOpt attacks over competing methods also remains highly similar to, if not the same as, the original results presented.
>
> 3. "The experiments lack some details: which is the decoder used for dimensionality reduction?"
> As stated in the first paragraph of Section 4.1, we adopt bilinear interpolation as the decoder, which is used in GenAttack (Alzantot et al., 2018) and Auto-ZOOM (Tu et al., 2018). This is to ensure fair comparison. However, the approach can be combined with different decoder types.

---

### Author Response · Authors · 2019-11-11
**Reply to all Reviewers**

We thank all the reviewers for their valuable comments and hope our responses address the issues raised. Following the reviewers’ suggestion, we have added results on 50 random CIFAR10 images (Response 2 to Reviewer#2, Table 2 and Figure 3 in our paper Section 5.2), and further verified the feasibility of our proposed BayesOpt attacks on ImageNet data (Response 2 to Reviewer#3, Section E in the Appendix of our paper).

Both Reviewer#2 and #3 find it surprising that Bayesian optimisation(BO) based on Gaussian process(GP) can handle such high-dimensional (d^r=14x14x3 even after dimensionality reduction) adversarial attack application. We’d like to take this space to elaborate our thoughts on this:

In general, there is no strong reason why GP-based BO cannot handle high dimensional problem. The performance of GPs depends on the kernel used and the complexity of the objective function. The 'high dimension' itself doesn't really mean much because for many applications, the effective dimension may be small (only a small number of input dimension have a significant impact on the objective function) (Chen et al., 2012; Wang et al.,2016b; Munteanu and Nayebi et al., 2019). The issue arises when there is complex dependencies among all the dimensions. Then simple kernels like RBF will fail while more complicated kernels such as deep neural network kernels (Wilson et al., 2016) will be costly to optimise and/or require more query data to train. The performances of GP-BOs in our paper, as well as in BO-ADMM(Zhao et al., 2019) kindly raised by Reviewer #1, suggest that the objective function in the adversarial problem, even though being very high-dimensional, probably doesn't depend on a complex combination of the input dimensions.

Moreover, the reasons why GP-BOs is undesirable for high-dimensional problem are usually:
    a) GP hyperparameter optimisation becomes high-dimensional and thus challenging, if we use automatic relevance determination(ARD). In our work, we turn off the ARD for our GP kernel. Although this compromises the expressiveness of our surrogate, it significantly ease the GP hyperparameter learning. As for our ADDGP-BO variant, we decompose the original search space of d^r=14x14x3 into 12 subspaces of d_s=49 and assign a different length-scale for each subspace. This contributes to more expressive modelling of the underlying function and thus high attack success rate than vanilla GP-BO.
    b) For high-dimensional search space, we usually require a large number of query data (N) to build an accurate model. Both of our contributions, ADDGP-BO attack and Bayesian learning of optimal reduced dimension, aim to alleviate this effect. By assuming additive structure $y =\sum_j^M y^j$ and decomposing the latent search space disjoint subspaces, we can model each subspace with a separate GP and better model the overall objective y with fewer query data. As for the online learning of optimal reduced dimension, it gives the chance to discover effective latent space of dimensions lower than 14x14x3 in the process of optimisation so that we can get a more accurate GP model given limited query number.
    c) The computation complexity of GPs scale cubically with N. In our paper, we focus on the attack query-efficiency and limit the query counts to a reasonable level of 1000, which standard GP can still handle. However, if we want to extend to the case with excessively large query budget, we need to resort to sparse GP methods to achieve the scalability. Moreover, we thank Reviewer #3 for the suggestion of using GPU acceleration (e.g. GPyTorch) to reduce the computation expense of GPs and sparse GPs.

Another possible reason why the GP-BO attack works is that the GP surrogate allows us to better exploit all the previous query information to infer the attack patterns. To briefly assess the quality of the GP surrogate, we actually explored another BO surrogate option, kernel density estimator/tree parzen estimators (Bergstra et al, 2013), which is supposed to better handle high-dimensional data and scales linearly with N. However, we found its performance is significantly worse than that of GP. We hypothesise that the good uncertainty estimation provided by GP is quite useful to finding the adversarial example.

Finally, it could be that the adversarial problem is actually a not-too-difficult optimisation problem as it doesn’t necessitate a global optimum to make a successful attack. In most of cases, a good local optimum is enough. As we can see that many gradient-based adversarial attacks can still get high attack success rates, despite that their gradient estimation may not be accurate and gradient descent can easily lead to a local optimum. Therefore, GP-BO, whose optimisation ability may be compromised by the high-dimensionality of the search space, is still able to find the successful adversarial perturbations.

---

### Decision · Program_Chairs · 2019-12-19

**Decision:**

Accept (Poster)

**Comment:**

This paper proposes a query-efficient black-box attack that uses Bayesian optimization in combination with Bayesian model selection to optimize over the adversarial perturbation and the optimal degree of search space dimension reduction. The method can achieve comparable success rates with 2-5 times fewer queries compared to previous state-of-the-art black-box attacks. The paper should be further improved in the final version (e.g., including more results on ImageNet data).